# Rapid and reversible epigenome editing by endogenous chromatin regulators

Simon M.G. Braun[1], Jacob G. Kirkland [1], Emma J. Chory [1,2], Dylan Husmann[1], Joseph P. Calarco[1] & Gerald R. Crabtree[1,3]

Understanding the causal link between epigenetic marks and gene regulation remains a central question in chromatin biology. To edit the epigenome we developed the FIRE-Cas9 system for rapid and reversible recruitment of endogenous chromatin regulators to specific genomic loci. We enhanced the dCas9–MS2 anchor for genome targeting with Fkbp/Frb dimerizing fusion proteins to allow chemical-induced proximity of a desired chromatin regulator. We find that mSWI/SNF (BAF) complex recruitment is sufficient to oppose Polycomb within minutes, leading to activation of bivalent gene transcription in mouse embryonic stem cells. Furthermore, Hp1/Suv39h1 heterochromatin complex recruitment to active promoters deposits H3K9me3 domains, resulting in gene silencing that can be reversed upon washout of the chemical dimerizer. This inducible recruitment strategy provides precise kinetic information to model epigenetic memory and plasticity. It is broadly applicable to mechanistic studies of chromatin in mammalian cells and is particularly suited to the analysis of endogenous multi-subunit chromatin regulator complexes.

---

[1] Departments of Pathology and Developmental Biology, Stanford University School of Medicine, Stanford, CA 94305, USA. [2] Department of Chemical Engineering, Stanford University, Stanford, CA 94305, USA. [3] Howard Hughes Medical Institute, Chevy Chase, MD 20815, USA. Simon M.G. Braun, and Jacob G. Kirkland contributed equally to this work. Correspondence and requests for materials should be addressed to G.R.C. (email: crabtree@stanford.edu)

Duuring development, epigenetic regulators coordinate gene expression changes that drive stem cell differentiation into different cell types. Epigenetic regulators modify chromatin compaction and DNA accessibility via multiple processes including post-translational histone modifications, DNA methylation, and nucleosome remodeling[1]. Recent human genome sequencing studies have called attention to the widespread role of chromatin regulators in human disease, often identifying unexpected biological roles for known chromatin regulators in human development[2–4]. These studies have revealed the highly cell type-specific nature of epigenetic regulation, underlining the need for new technologies to study the function of chromatin regulators in specific cell types, at specific developmental times and in their proper genomic contexts. Methods using in vitro chromatin templates have not reflected these new

discoveries. For example, BAF250a and BAF250b, two mutually exclusive mSWI/SNF (BAF) complex subunits have very different mutation patterns in human disease. BAF250a is mutated in many human malignancies[3], whereas BAF250b is the most commonly identified de novo mutated gene in human neurodevelopmental disorders[5], yet neither of these subunits are required for the in vitro activities of the BAF chromatin-remodeling complex. New methods are essential to understand the mechanisms by which these essential epigenetic regulators carry out their distinct biologic roles in different cell types.

Understanding the causal link between epigenetic marks and gene expression remains a central question in chromatin biology especially as recent advances in epigenome editing techniques are beginning to shed new light on these processes. The discovery of CRISPR-Cas9 interference, by targeting a catalytically dead

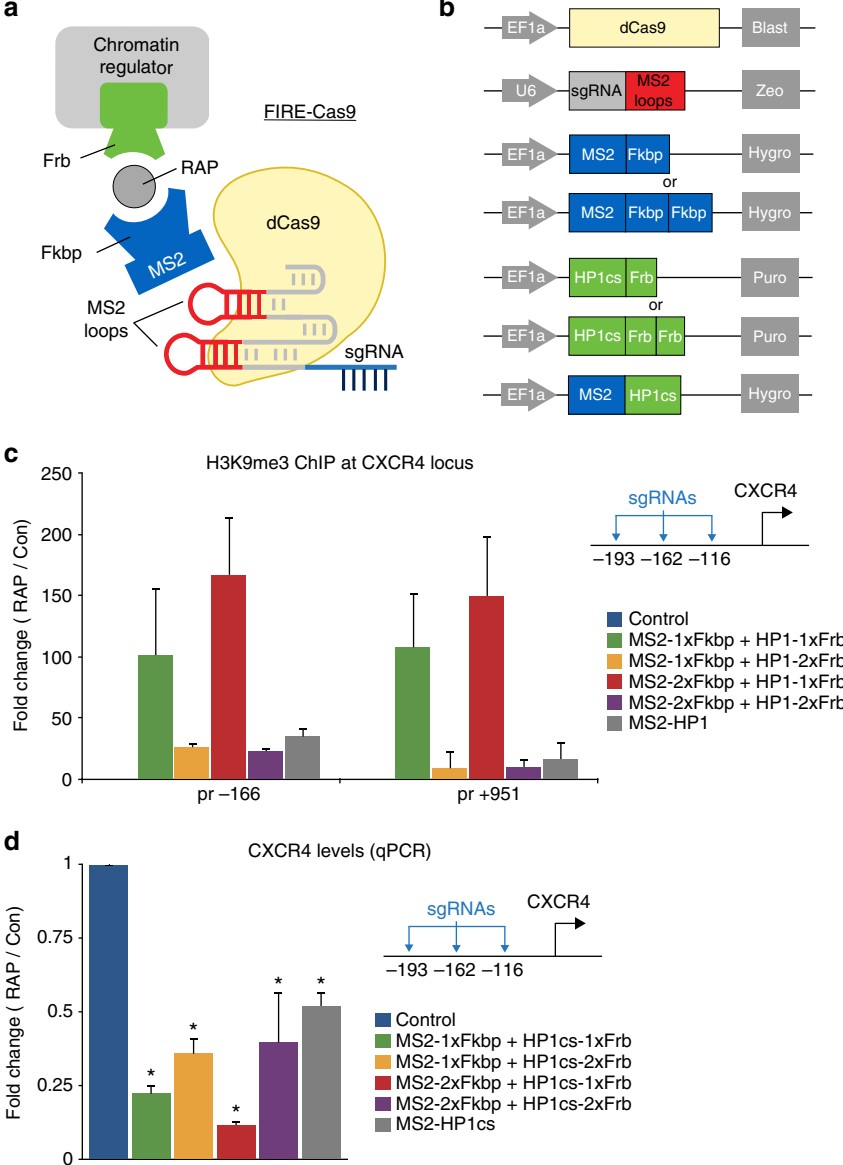

**Fig. 1** Inducible heterochromatin complex recruitment silences target gene expression in HEK 293 cells. **a** Schematic representation for rapid and reversible recruitment of chromatin regulators to specific genomic loci by FIRE-Cas9. **b** Lentiviral constructs used for dCas9, sgRNA-MS2, MS2-Fkbp (1x or 2x) and HP1cs-Frb (1x or 2x), or MS2-HP1cs expression in HEK 293 cells. Lentiviral integration is selected for with four unique resistance genes. **c** H3K9me3 ChIP analysis at the *CXCR4* locus (−166 bp and +951 bp from TSS) after 5 days of RAP treatment to induce HP1cs recruitment. Fold changes represent RAP/Control (Con: untreated, no RAP) in respective cell lines except for direct fusion experiment where fold change represents MS2-HP1/MS2-Fkbp recruitment (both untreated, no RAP). *n* = 3; error bars s.e.m. **d** *CXCR4* expression levels measured by qPCR after 5 days of RAP treatment to recruit HP1cs. *n* = 3; error bars s.e.m. *p* < 0.05

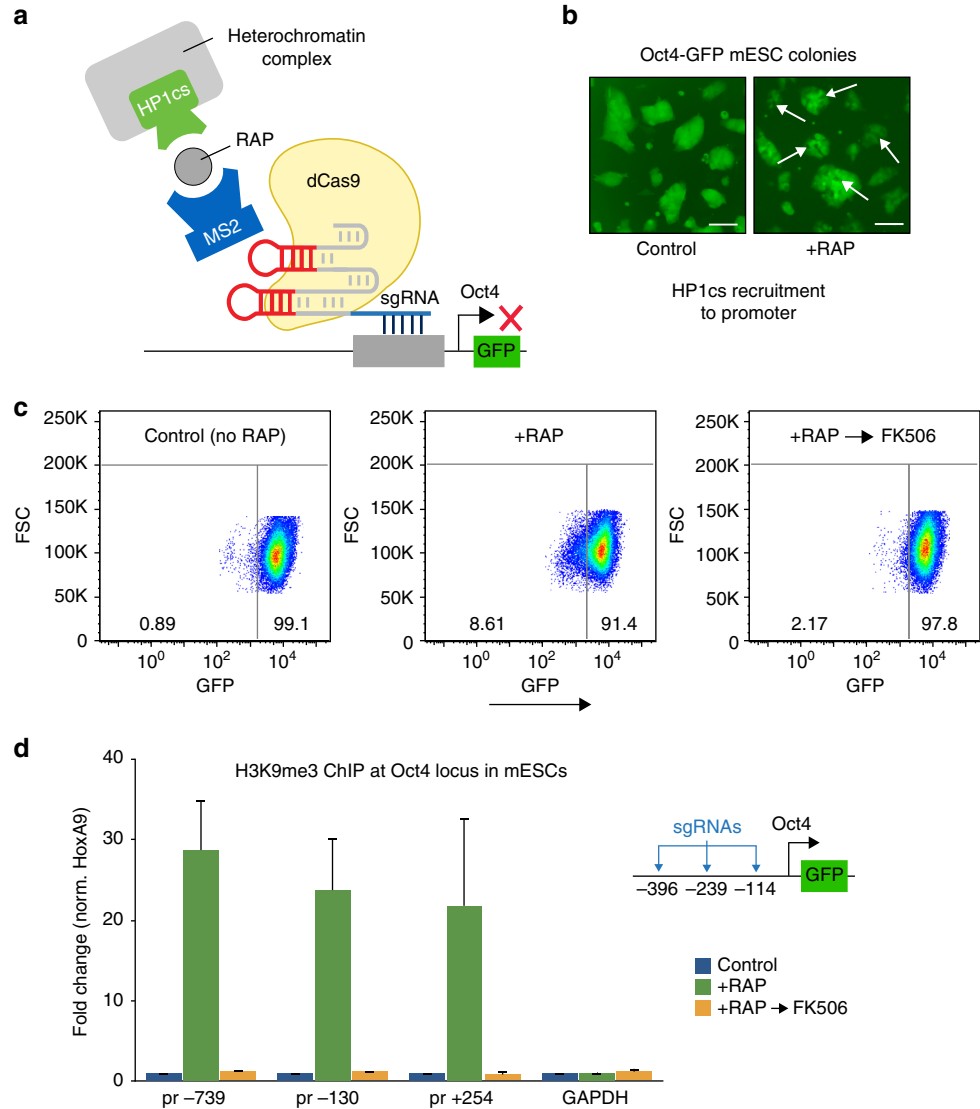

**Fig. 2** Reversible heterochromatin complex recruitment regulates Oct4 expression in mESCs. **a** Schematic representation of heterochromatin complex recruitment to the *Oct4* locus in reporter Oct4-GFP mESCs. **b** Microscopy reveals loss of GFP expression in Oct4-GFP mESC colonies 5 days after RAP treatment to induce HP1cs recruitment. *Arrows* show examples of GFP-negative cells within Oct4-GFP ESC colonies. *Scale bar* represents 100 µm. **c** FACS analysis shows reversible repression of Oct4-GFP expression in mESCs. After 5 days of RAP to recruit HP1cs to the *Oct4* locus, washout experiments using FK506 restore GFP expression levels after 5 days. Numbers represent % of GFP±cells **d** H3K9me3 ChIP analysis at the *Oct4* locus (−739, −130, and +254 bp from transcriptional start site, TSS) 5 days post RAP treatment and 5 days after washout with FK506. n = 3; *error bars* s.e.m

mutant of *Streptococcus pyogenes* SpCas9 (dCas9) to block transcription, has provided a valuable tool for regulating gene expression[6, 7]. Several groups have since fused dCas9 to well-characterized repressors and activators (e.g., KRAB and VP64) to modulate gene expression with enhanced silencing and activation capacity[8, 9]. Furthermore, novel tagging approaches have allowed more efficient recruitment of multiple effectors to a single-dCas9 anchor bound to a specific genomic locus[10, 11]. Recruitment strategies have also been combined with chemically inducible approaches to provide temporal control of transcriptional regulation[12, 13]. Finally, recent studies have also focused on regulatory DNA sequences, via the recruitment of dCas9 fused to the histone acetyl-transferase p300 or dCas9 fused to the DNA demethylase Tet1 to activate enhancers[14, 15]. While these technologies provide new methods for epigenome editing, they focus on the recruitment of synthetic modulators and lack the temporal resolution and reversibility required for mechanistic studies of epigenetic regulation.

In this study we describe a method, Fkbp/Frb inducible recruitment for epigenome editing by Cas9 (FIRE–Cas9), which allows rapid and reversible recruitment of endogenous chromatin complexes to any genomic locus in almost any cell type. Many of the enzymes that are responsible for writing, erasing, and reading epigenetic marks are present in multi protein complexes that bind chromatin. Previously described methods recruit exogenous activators/repressors to turn gene expression on and off in cells cultured over several days. In contrast, this endogenous complex recruitment approach uses induced proximity[16] which enables us to determine causal links between epigenetic regulators and histone modifications within minutes of recruitment. By fusing a single subunit of a chromatin complex with a chemical-induced proximity tag, Frb (FKBP-rapamycin-binding domain of mTOR), we can rapidly recruit intact multi-subunit complexes to a specific genomic sequence upon rapamycin (RAP) treatment as described originally for signaling proteins[16]. Locus specificity is obtained via expression of a complementary dimerizer Fkbp (FK506-binding-protein) that

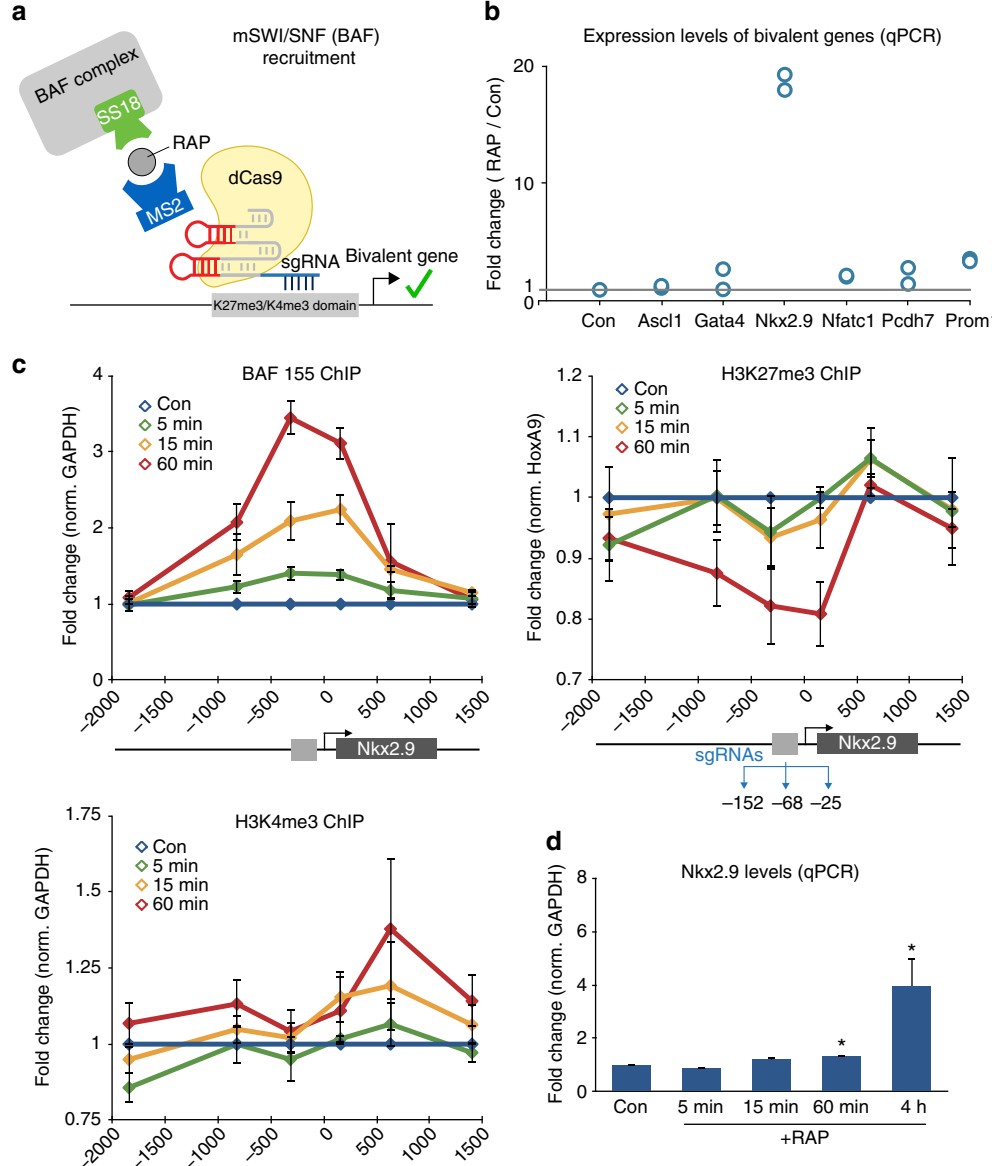

**Fig. 3** Rapid BAF complex recruitment to the bivalent *Nkx2.9* locus activates transcription in mESCs. **a** Schematic representation of rapid BAF complex recruitment to bivalent loci in mESCs with SS18-Frb. **b** Using specific MS2-sgRNAs we targeted the BAF complex to 6 different bivalent loci in mESC lines. After 48 h RAP treatment, we detected a 2-fold to 3-fold increase in *Gata4*, *Nfatc1*, *Pcdh7*, and *Prom1* gene expression levels, as well as a 20-fold increase in *Nkx2.9* expression by qPCR. *n* = 2 per condition **c** Rapid BAF recruitment studies by RAP treatment performed over minutes at the *Nkx2.9* locus. BAF155 ChIP shows enrichment for this subunit of the complex after 5 min of SS18-Frb recruitment. ChIP experiments show loss of H3K27me3 over the bivalent recruitment site after 60 min RAP treatment, whereas H3K4me3 levels increase over the gene body. *n* = 3; *error bars* s.e.m. **d** *Nkx2.9* expression levels measured by qPCR during the time course of BAF recruitment. *n* = 4; *error bars* s.e.m. *p* < 0.01

is fused to a dCas9–MS2 anchor (Fig. 1a). While this strategy is broadly applicable to many chromatin regulators, in this study we focused on the recruitment of Hp1/Suv39h1 heterochromatin complex as well as the BAF chromatin-remodeling complex to utilize this tool in the context of both gene repression and activation. The recruitment studies presented here provide new insight into the finely tuned epigenetic mechanisms that determine transcriptional output in mammalian cells.

## Results

**Inducible HP1 complex recruitment silences gene expression.** To determine the optimal inducible recruitment strategy we first focused on heterochromatin complex recruitment in human embryonic kidney (HEK 293) cells. In mammalian cells, the

histone H3 lysine 9 (H3K9) methyltransferase Suv39h1 forms a complex with the heterochromatin protein 1 (HP1) as well as other methyltransferases (e.g., SetDB1, G9a) and is responsible for the deposition and propagation of the heterochromatic H3K9 tri-methylation (me3) modification[17]. HP1 interacts with Suv39h1 through a C-terminal chromo-shadow (cs) domain and binds H3K9me3-modified histones via an N-terminal chromodomain[18]. To target this repressive complex to chromatin, we fused the HP1cs domain to a Frb tag[19] and expressed the fusion protein in HEK 293 cells using lentiviral transduction (Fig. 1b and Supplementary Fig. 1a). Together with the Frb fusion proteins, we expressed dCas9 as well as sgRNAs targeting three loci upstream of the highly expressed *CXCR4* (C-X-C motif chemokine receptor 4) gene (−193, −162, and −116 bp). We selected the sequences based on previously validated sgRNAs for this locus[9]. The sgRNAs

used were fused to 2x MS2 RNA hairpin loops, allowing for recruitment of four MS2 bacteriophage coat proteins to each dCas9 anchor, which has been shown to be the most efficient platform for multiple effector recruitment[9]. To allow for inducible recruitment, we expressed MS2-Fkbp fusion proteins in these cells. Treatment of cells with 3 nM RAP-induced Fkbp/Frb dimerization and subsequent recruitment of the heterochromatin complex. At this low concentration, RAP treatment did not inhibit cell proliferation. Using chromatin immunoprecipitation (ChIP) we detected H3K9me3 deposition at the recruitment site as well as 1 kb upstream (Fig. 1c). Therefore HP1cs recruitment by FIRE–Cas9 is sufficient to deposit the H3K9me3 repressive mark as well as promote the propagation of the mark. We tested all different combinations of 1x or 2x Fkbp and Frb tags (Fig. 1b and Supplementary Fig. 1a) and found that optimal recruitment was detected using MS2-2xFkbp and HP1cs-1xFrb. We also performed direct recruitment experiments via MS2-HP1cs fusion proteins but detected lower levels of H3K9me3 compared to the Fkbp/Frb dimerizing system. The increased recruitment using MS2-2xFkbp and HP1cs-1xFrb dimerization is likely due to a clouding effect mediated by the rapid on/off kinetics of the dimerizing system (we estimated the mean residence time of the recruited complex on chromatin to be ~83 s[20, 21]). The clouding results in an increased local concentration of recruited Frb-fusion proteins upon RAP treatment. Therefore we show that this inducible approach leads to more robust recruitment and histone modifications than classic direct fusion approaches. To evaluate the gene silencing ability of HP1 complex recruitment, we measured CXCR4 gene expression levels by qPCR. CXCR4 mRNA levels were decreased by up to 90% when using MS2-2xFkbp and HP1cs-1xFrb recruitment (Fig. 1d). These results reveal a positive correlation between H3K9me3 levels and silencing efficiency when comparing the different recruitment approaches. To control for effects of targeting dCas9 to the CXCR4 locus, we measured CXCR4 levels in dCas9 expressing HEK 293 cells with and without sgRNAs for all conditions, and found no changes in expression levels (Supplementary Fig. 1b). To confirm the efficiency of H3K9me3-mediated gene silencing we generated lines recruiting HP1cs to weakly and intermediate expressed genes, ASCL1 (achaete-scute family bHLH transcription factor 1) and NEUROD1 (neuronal differentiation 1), respectively, in HEK 293 cells and observed similar reductions in gene expression by qPCR (Supplementary Fig. 1c). In addition, this inducible system can also be used to activate gene expression upon RAP treatment via recruitment of VP64–Frb fusion proteins (Supplementary Fig. 1d).

**Reversible HP1 complex recruitment regulates Oct4 levels**. Next we tested the efficiency and reversibility of heterochromatin complex recruitment in a different mammalian cell type, mouse embryonic stem cells (mESCs), a model cell line widely used to study epigenetic mechanisms. Using lentiviral expression we expressed dCas9, MS2-2xFkbp and HP1cs-Frb fusion proteins as well MS2-sgRNAs targeting the regulatory sequences upstream of the Oct4 (Pou5f1, POU class 5 homeobox 1) gene (−114, −293, and −396 bp) in reporter Oct4-GFP mESCs[22] (Fig. 2a). In embryonic stem cells, the Oct4 transcription factor is highly expressed and is required for pluripotency and self-renewal[23]. During ES cell differentiation, Oct4 expression is rapidly silenced through a series of events including histone H3K9 methylation[24]. Upon treatment of the transduced Oct4-GFP ESCs with RAP we observed a decrease in reporter GFP signal, suggesting that HP1cs recruitment represses Oct4 expression. The addition of FK506, a dimeric competitive inhibitor of RAP that binds only to Fkbp thus blocking Fkbp/Frb dimerization, allows for the reversal of chromatin regulator recruitment. Importantly, following washout with FK506, GFP

levels returned to normal (Fig. 2b, c) and H3K9me3 ChIP showed that the repressive mark that was placed over the recruitment site upon RAP addition was erased (Fig. 2d). Together these results demonstrate the reversibility of the FIRE–Cas9 system in mESCs, an essential characteristic for mechanistic studies of epigenetic regulation aiming to reveal causal links between chromatin modifications and gene expression.

**BAF recruitment to bivalent genes activates transcription**. To determine how widely applicable the FIRE–Cas9 approach might be, we sought to activate gene expression by recruiting the multi-subunit mSWI/SNF (BAF) chromatin regulatory complex. These large 2MDa complexes are combinatorially assembled into tissue-specific forms with at least 15 subunits encoded by 28 different genes and use energy provided by ATP hydrolysis to remodel chromatin[25, 26]. Recent studies have shown that BAF complexes oppose Polycomb by antagonizing PRC1/2 complexes that deposit and bind to the repressive H3K27me3 and H2Aub histone marks[27–29]; however, the BAF complex has yet to been shown to be sufficient for activating gene expression. To recruit BAF we fused the SS18 subunit to an N-terminal Frb tag, as previous work from our lab has shown that this fusion protein incorporates into a functional and recruitable BAF complex[29] (Fig. 3a and Supplementary Fig. 2a). We then designed multiple MS2-sgRNAs to screen different bivalent promoters in mESCs characterized by active (H3K4me3) and repressive (H3K27me3) marks that are thought to poise genes for transcription (Supplementary Fig. 2b). This allows us to test if dCas9–MS2 mediated BAF recruitment is sufficient to oppose H3K27me3 through its antagonism with PRC2 complexes, leading to bivalent gene activation. We screened 6 bivalent genes that are known targets of BAF in differentiated cells but lack Brg1 (ATPase subunit of the complex) peaks in mESCs as determined by ChIP-seq[30]. We observed modest 2-fold to 3-fold increases in expression at four bivalent genes (Gata4, Nfatc1, Pcdh7, and Prom1). Remarkably, we detected a 20-fold induction in transcription of the Nkx2.9 gene following BAF recruitment to its bivalent promoter (Fig. 3b and Supplementary Fig. 2c). These results reveal that BAF recruitment is sufficient to induce transcription at specific bivalent loci, which perhaps are already primed developmentally. To study the mechanism of BAF-dependent induction of bivalent gene transcription we performed ChIP analyses at 24 and 72 h post RAP induction at Nkx2.9. We found that SS18 recruitment led to an increase in the BAF155 subunit over the recruitment site upon RAP treatment, suggesting the multi-subunit complex is being recruited to the bivalent locus (Supplementary Fig. 3b). Furthermore we saw a loss in H3K27me3 and an increase in H3K4me3 levels over the Nkx2.9 locus, coupled with a 20-fold increase in Nkx2.9 mRNA levels at 48 h (Supplementary Fig. 3a, c). These results show that BAF recruitment to the Nkx2.9 locus for a period of days leads to chromatin and transcriptional changes, but the possibility that this is due to indirect downstream effects remained. To determine if these changes are direct consequences of BAF recruitment, we performed recruitment experiments over shorter time courses (5, 15, and 60 min). We detected increases in BAF155 levels as early as 5 min post RAP treatment and these levels increased gradually up to 60 min. Notably, we also measured a rapid loss in H3K27me3 levels after BAF recruitment coupled with an increase in H3K4me3 levels (Fig. 3c) during the 60 min RAP time course. However, transcription of the Nkx2.9 gene was only modestly induced at 60 min (~1.5-fold over no recruitment) with larger increases in transcription only seen after 4 h of RAP treatment (~5-fold) (Fig. 3d). This kinetic analysis reveals that the rapid epigenetic events precede transcription. Therefore, for the first time, we demonstrate that targeting the BAF complex to bivalent genes can induce transcription through loss of H3K27me3 and increases in H3K4me3.

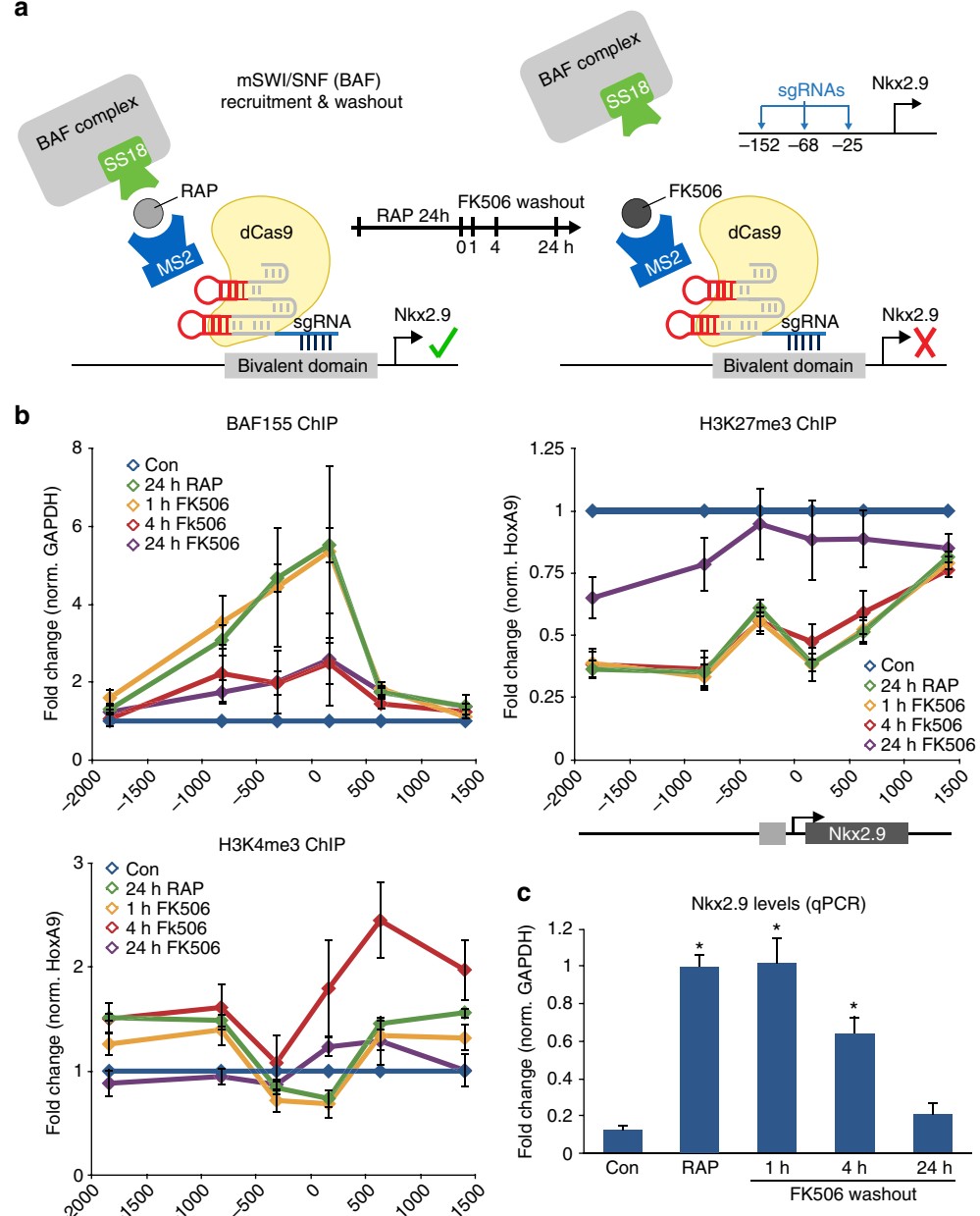

**Fig. 4** Transient BAF-dependent activation of *Nkx2.9* does not form a stable epigenetic memory **a** Schematic representation of BAF complex recruitment with SS18-Frb (+RAP) and subsequent removal (+FK506) at the *Nkx2.9* locus in mESCs **b** Rapid BAF washout studies by FK506 treatment performed for 1, 4, and 24 h post 24 h recruitment (+RAP) to the *Nkx2.9* locus. BAF155 ChIP shows a loss of this subunit of the complex over the recruitment site after 4 h of SS18-Frb washout. ChIP experiments show an increase in H3K27me3 levels over the bivalent domain after 24 h FK506 treatment, whereas H3K4me3 levels decrease. *n* = 4; *error bars* s.e.m. **c** *Nkx2.9* expression levels measured by qPCR during the time course of BAF removal. *n* = 4; *error bars* s.e.m. *p* < 0.01

**BAF-induced gene activation is not a stable epigenetic state.** Next, we evaluated the consequence of BAF removal from this newly activated bivalent locus. We performed a time course using FK506 to rapidly washout SS18-Frb following 24 h of BAF recruitment at the *Nkx2.9* locus (Fig. 4a). After 4 h of FK506 treatment we measured a loss in BAF155 levels over the recruitment site as well as increases in H3K27me3 and H3K4me3 levels (Fig. 4b). These epigenetic changes coincided with decreased levels of *Nkx2.9* expression (Fig. 4c). After 24 h of BAF removal from the bivalent locus the epigenetic landscape returned to control conditions as H3K27me3 levels increased and H3K4me3 decreased, resulting in silencing of *Nkx2.9* expression. Thus, following BAF washout from this locus, the bivalent nature of the epigenetic domain is reset as Polycomb returns and

transcription is switched off once again. Transient BAF recruitment is not sufficient to maintain long-term expression at bivalent genes in mESCs. These results highlight the strengths of the FIRE–Cas9 recruitment strategy, namely the flexibility to screen multiple loci across the genome as well as interrogate rapid kinetic changes in chromatin landscapes.

**Discussion**

Here we describe a method for rapid and reversible epigenome editing by endogenous chromatin regulators called FIRE–Cas9. We show that chemical dimerizers or inducers of proximity can be used to recruit the Hp1/Suv39h1 heterochromatin complex as well as the mSWI/SNF (BAF) chromatin-remodeling complex to

specific genomic loci by dCas9 in mammalian cells. The recruitment of these repressive and activating complexes to chromatin modifies local histone marks that in turn regulate gene expression levels. HP1 complex recruitment to a specific gene leads to H3K9me3 deposition, which then spreads over the locus to repress gene expression after 5 days. Washout experiments showed that recruitment could be reversed with a competitive inhibitor of RAP, leading to H3K9me3 loss and restored gene expression. Therefore, H3K9me3 marks are not permanently inherited when deposited at a highly expressed gene in mESCs as previously described[19, 31]. BAF complex recruitment to a bivalent gene rapidly evicts H3K27me3 marks through opposition to Polycomb repressive complexes within minutes, resulting in increased H3K4me3 levels and subsequent gene activation hours after the epigenetic changes. Furthermore, washout studies reveal that transient BAF-dependent activation of bivalent genes by Polycomb eviction does not form a stable epigenetic memory, as the chromatin landscape is reset within hours rather than stably inherited, and transcription is silenced. The reassembly of facultative heterochromatin after RAP washout provides a novel assay for studying inducible heterochromatin assembly at many loci over the genome. Thus, by recruiting large multi-subunit complexes and studying their influence on chromatin dynamics over timescales ranging from minutes to days, we can study and order the causal steps between epigenetic mechanisms and transcriptional regulation.

Using this recruitment method, we are able to screen multiple bivalent loci for BAF-induced transcription, by harnessing the ability of dCas9 to target many loci through the expression of specific sgRNA libraries. This high throughput approach is faster than traditional epigenome editing methods that require knock-ins of protein recognition DNA elements such as Gal4 or Zinc-finger arrays[19]. Despite all loci being characterized by similar epigenetic landscapes in mESCs only the *Nkx2.9* gene, which is normally expressed during neural progenitor differentiation into motor neurons[32], was strongly induced by BAF recruitment. Future studies are required to determine the co-regulators present at this locus that allow induction of transcription by a chromatin-remodeling complex. For example, genetic screens to identify novel regulators of BAF function could be performed using an Nkx2.9 reporter as a readout for BAF mediated gene activation. Furthermore, this technique can be applied to many different protein complexes that bind chromatin in a variety of different mammalian cell types that can be efficiently transduced with lentiviruses. Mass spectrometry analyses of certain chromatin complexes such as the BAF complex and the PRC1 complex have identified multiple subtypes of these complexes made up of core and unique sub-units[33–35]. The method can be used to recruit canonical and non-canonical forms of the same complex to relevant chromatin loci using Frb fusions to specific subunits to study variant specific molecular functions. Also, exome-sequencing studies have revealed that subunits of the mSWI/SNF complex are mutated in over 20% of human cancers[3], suggesting that recruitment studies using endogenous mutant complexes in cancer cells may reveal novel disease mechanisms. Finally, widespread roles for chromatin regulators have been discovered by exome-sequencing studies in many human diseases, often characterized by unexpected tissue-specific or developmentally specific mechanisms. Our approach allows one to use an appropriate cell type and well-defined target genes to understand the mechanism by which these chromatin regulators function in vivo.

## Methods

**Cell culture**. HEK 293 cells (ATCC) were cultured using standard conditions in media containing: DMEM (Life Technologies), 10% FBS (Applied StemCell), and Penicillin-Streptomycin (Life Technologies).

ES cells were cultured using standard conditions in media containing: DMEM (Life Technologies), 7.5% ES-sure FBS (Applied StemCell), 7.5% KnockOut SR (Life Technologies) Penicillin-Streptomycin (Life Technologies), Glutamax (Life Technologies), HEPES buffer (Life Technologies), 2-mercaptoethanol (Life Technologies), and MEM-NEA (Life Technologies). LIF was replaced daily and ES cells were passaged every 48 h.

For long recruitment experiments (>1 h), cells were treated with 3 nM RAP (Selleckchem) and/or 3 nM FK506 (Abcam) added directly to culture dish and changed daily. For short recruitment experiments (≤1 h), cells were treated with 30 nM RAP and/or FK506 in suspension.

**Lentivirus production**. HEK 293T cells were transfected with lentiviral constructs and packaging plasmids (pspax2 and pMD2.G) using PEI transfection (Polysciences). Two days post transfection the virus containing cell culture media was collected, filtered with a 0.4 μm Steritop filter (Millipore), and centrifuged at $50,000 \times g$ for 2 h at 4 °C (SW28 rotor on ultracentrifuge). The viral pellet was resuspended in PBS and used for subsequent infections. Selection of lentiviral constructs was achieved with: puromycin (1.5 μg per mL), blasticidin (10 μg per mL), hygromycin (200 μg per mL), and zeocin (200 μg per mL).

**Chromatin immunoprecipitation**. ChIP experiments were performed as previously described by Hathaway et al.[19]. In brief, cultured cells were trypsinized for 5 min, washed with PBS, and fixed for 12 min by addition of formaldehyde to a final concentration of 1%. Crosslinking was then quenched with 0.125 M glycine and cells were incubated on ice for 5 min. Crosslinked cells were spun at $800 \times g$ for 5 min. Nuclei were prepared with 10 mL cell lysis buffer (50 mM HEPES pH 8.0; 140 mM NaCl; 1 mM EDTA; 10% glycerol; 0.5% NP40; 0.25% Triton × 100), then washed in 10 mL rinse buffer (10 mM Tris pH 8.0; 1 mM EDTA; 0.5 mM EGTA; 200 mM NaCl). The chromatin pellets were resuspended in 900 μL shearing buffer (0.1% SDS, 1 mM EDTA pH 8.0, and 10 mM Tris pH 8.0) and transferred to a Covaris tube for 12 min of sonication using a Covaris focused ultrasonicator at 5% duty cycle, intensity 4, 140 PIP, and 200 cycles per burst. Sonicated chromatin was spun at $10,000 \times g$ for 5 min, and the supernatant was collected.

The IP reactions were setup as follows: the sonicated chromatin was diluted with 0.25 volume of 5×IP buffer (250 mM HEPES, 1.5 M NaCl, 5 mM EDTA pH 8.0, 5% Triton X-100, 0.5% DOC, and 0.5% SDS) and incubated for 12–16 h at 4 °C with 25 μL protein G Dynabeads (Life Technologies) and 5 μg of antibody. The beads were then washed three times with 1 mL 1×IP buffer, once with 1 mL DOC buffer (10 mM Tris pH 8; 0.25 M LiCl; 0.5% NP40; 0.5% DOC; 1 mM EDTA), once with 1 mL TE buffer and then eluted in 300 μL elution buffer (1% SDS0.1 M NaHCO₃). DNA was purified using NuceloSpin clean-up columns (Machery Nagel) and resuspended in 30 μL EB solution.

Antibodies (5 μg/IP): H3K9me3 (#ab8898, Abcam); BAF155 (Crabtree lab); H3K27me3 (#39155, Active Motif); H3K4me3 (#05-745R, EMD Millipore).

**qPCR**. For RT-qPCR analysis, RNA was extracted from cells using Trisure (Bioline) and cDNA was synthesized from 1 μg RNA using the SensiFAST kit (Bioline).

For ChIP-qPCR and RT-qPCR, samples were prepared using the SensiFAST SYBR Lo-Rox kit (Bioline, BIO-94020), according to the manufacturer's instructions. Analysis of qPCR samples was performed on a QuantStudio 6 Flex system (Life Technologies). For ChIP-qPCR experiments, enrichment (bound over input) values were normalized to values with no RAP treatment (RAP/Con). Student's two-sample unpaired *t*-tests were performed to determine statistical significance.

**Flow cytometry analysis**. FACS experiments were performed on a Scanford analyzer and data analysis was performed using FlowJo software. Individual ES cells were gated based on forward and side scatter, Propidum Iodide positive cells were omitted from the analysis, and remaining cells were analyzed for GFP levels.

**Western blot**. Cells were lysed in RIPA buffer and protein concentrations were determined by Bradford assay (Biorad). Proteins were separated by SDS–PAGE electrophoresis with a 4–12% Bis-Tris protein gel (Thermo Scientific) and then transferred to an Immobilon-FL membrane (Millipore). Blots were probed with primary antibodies, rabbit α-HA (1:1000, Abcam #ab9110), rabbit α-FKBP (1:1000, Abcam #ab2918), rabbit α-FRB (1:1000, Crabtree lab) and mouse α-GAPDH (1:2000, Santa Cruz #sc32233) followed by fluorescence-conjugated secondary antibodies (Li-Cor) and bands were detected using an Odyssey CLX imaging system (Li-Cor).

**Plasmids and primers**. The lentiviral plasmid constructs were modified from Konermann et al.[10] and Hathaway et al.[19]:

Lv EF1a dCas9 2A Blast; Lv U6 sgRNA-2xMS2-RNA EF1 Zeo; Lv EF1a MS2-Fkbp(1x or 2x) 2A Hygro; Lv EF1a MS2-HP1cs 2A Hygro; Lv EF1a HP1cs-Frb(1x or 2x) PGK Puro; Lv EF1a SS18-Frb PGK Puro.

The sgRNAs used for targeting *CXCR4*, *ASCL1*, and *NEUROD1* loci were previously validated by Chavez et al.[9]. Other sgRNAs sequences were designed using the tool provided at crispr.mit.edu (Supplementary Table 1).

ChIP-qPCR and RT-qPCR primer sequences are listed in Supplementary Tables 2 and 3. Detailed plasmid sequence information can be found in Supplementary Note 1[36–38].

**Data availability**. All data and reagents generated during this study are included in this published article and available from the corresponding author upon request.

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

## Acknowledgements

This manuscript is dedicated to the memory of Joe P. Calarco, a brilliant scientist, dedicated colleague, and true friend. We thank the Crabtree lab, C. Weber, K. Majzoub, R. Kamakaka and J.A. Calarco for helpful discussions and E. Miller for sharing curated GEO data sets. Lentiviral plasmids were a gift from N. Hathaway, C. Kadoch, F. Zhang, and D. Trono (via Addgene). FACS analysis was performed using the Stanford Shared FACS Facility. This work was supported by an NIH grant. G.R.C. is an HHMI Investigator. S.M.G.B. is supported by a Swiss National Science Foundation postdoctoral fellowship (SNSF-P300PA164675). J.G.K. is supported by a USAMRAA grant (W81XWH-16-1-0083) and a National Cancer Institute grant (T32 CA09151). E.J.C. is supported by an NIH grant (5F31CA203228-02).

## Author contributions

S.M.G.B. and J.G.K. designed and conducted the experiments, analyzed data and wrote the manuscript. E.J.C. and D.H. performed experiments. J.P.C. designed experiments. G.R.C. designed experiments and wrote the manuscript.

## Additional information

**Competing interests:** The technology described in this manuscript has been patented.

