## [Peer Review File · Nature Communications]

Reviewers' Comments:

Reviewer #1:

Remarks to the Author:

NCOMMS-17-14356-T

“Rapid and reversible epigenome editing by endogenous chromatin regulators”

Braun et al. developed a CRISPR/Cas9-based system for inducible recruitment of chromatin regulators to specific genomic loci. They combined dCas9/MS2 and Fkbp/Frb technologies to make an inducible system for epigenome editing. They recruited Hp1/Suv39h1 heterochromatin complex and mSWI/SNF (BAF) chromatin remodeling complex by Frb-fused HP1cs domain and SS18, respectively.

Similar inducible epigenome editing system has been recently reported (Gao Y et al. 2016, PMID: 27776111). Several epigenome editing systems recruiting several chromatin modifiers have been reported. Recently, authors reported a similar system using ZFHD1 zinc finger and Fkbp/Frb which recruiting mSWI/SNF (BAF) chromatin remodeling complex (Kadoch et al. 2017, PMID: 27941796) to ZFHD1-recognition sequences knock-in sites. However, the current paper is the first report of CRISPR/Cas9-based system for inducible recruitment of chromatin regulators to specific genomic loci. Therefore, this method could be used by the community for easy manipulation of chromatin states.

Several points raised were improved however following point is not improved and critical for publication.

Comments:

7. The information enough for reproduction of the experiments should be included in methods or in supplemental figures. Author did not describe full sequence information for the plasmids used. It is critical for the reproduction of the experiments.

Reviewer #2:

Remarks to the Author:

The authors have addressed my previous concerns appropriately. Furthermore, the new experiment has greatly increased the impact of this work and utility of this approach, I am very enthusiastic about its publication in Nat Com.

Reviewer #2 (Remarks to the Author):

“... However, the current paper is the first report of CRISPR/Cas9-based system for inducible recruitment of chromatin regulators to specific genomic loci. Therefore, this method could be used by the community for easy manipulation of chromatin states.

Several points raised were improved however following point is not improved and critical for publication.

Comments:

7. The information enough for reproduction of the experiments should be included in methods or in supplemental figures. Author did not describe full sequence information for the plasmids used. It is critical for the reproduction of the experiments.”

We appreciate the reviewer’s comments on the broad applications for our inducible epigenome editing strategy and its use to the chromatin field. To address this last point we have now included in the supplementary information the full sequences of the different plasmids we used in this study. In addition, we plan to deposit these plasmids in the Addgene database, where we will also include the full sequence information to make these reagents available to labs in the chromatin field.

Reviewer #4 (Remarks to the Author):

“The authors have addressed my previous concerns appropriately. Furthermore, the new experiment has greatly increased the impact of this work and utility of this approach, I am very enthusiastic about its publication in Nat Com.”

We thank the reviewer for her/his appreciation of our inducible epigenome editing strategy and for their insightful comments that have helped improve our manuscript.